# Working Memory Ability Evaluation Based on Fuzzy Support Vector Regression

**DOI:** 10.3390/s23198246

**Published:** 2023-10-04

**Authors:** Jia-Hsun Lo, Han-Pang Huang, Su-Ching Sung

**Affiliations:** 1Department of Mechanical Engineering, National Taiwan University, Taipei 10617, Taiwan; 2Department of Gerontology and Health Care Management, Chang Gung University of Science and Technology, Taoyuan City 33303, Taiwan; scsung@mail.cgust.edu.tw

**Keywords:** working memory, EEG, fuzzy SVR, cognition

## Abstract

One’s working memory process is a fundamental cognitive activity which often serves as an indicator of brain disease and cognitive impairment. In this research, the approach to evaluate working memory ability by means of electroencephalography (EEG) analysis was proposed. The result shows that the EEG signals of subjects share some characteristics when performing working memory tasks. Through correlation analysis, a working memory model describes the changes in EEG signals within alpha, beta and gamma waves, which shows an inverse tendency compared to Zen meditation. The working memory ability of subjects can be predicted using multi-linear support vector regression (SVR) with fuzzy C-mean (FCM) clustering and knowledge-based fuzzy support vector regression (FSVR), which reaches the mean square error of 0.6 in our collected data. The latter, designed based on the working memory model, achieves the best performance. The research provides the insight of the working memory process from the EEG aspect to become an example of cognitive function analysis and prediction.

## 1. Introduction

Cognition is an important part of human brain activity which affects our thinking and behavior, and cognitive impairment is often related to many brain diseases such as Alzheimer’s disease. Electroencephalography (EEG) researchers can acquire the signals of brain activity, so-called “brain waves”, for further analysis. Many brain states, such as emotion and cognitive load, have been discovered in previous research. A study in 2016 estimated the cognitive load using EEG connectivity measures [1]. The model designed using regularized linear discriminant analysis (RLDA) classified cognition load into two categories, which reached an accuracy of 60%. Despite the performance needing improvements, several approaches were proposed to measure connectivity in the brain region. With the spread of machine learning applications, a study constructed the classifier using recurrent neural network (RNN) to deal with four categories of cognition load with an accuracy of up to 92.5% in 2018 [2]. The RNN classifier consists of the convolutional neural network (CNN) and long-short term memory (LSTM). The machine learning classifier shows high performance but lacks the ability to explain the mechanisms of brain activity. In 2020, the approach of cognition load analysis was applied to scenarios in driving tasks with multi-type data analysis including EEG signals [3]. The classifier was trained using machine learning algorithms such as support vector machine (SVM), k-nearest neighbors (KNN) and random forest. Dealing with three categories, the F-score was about 0.8, proving that the brain state can be acquired through multi-type measurements. SVM and KNN were also utilized to remove artifacts [4]. Another study found the EEG temporal features using bidirectional long-short term memory (BiLSTM) in emotion recognition [5], which satisfied the frontal EEG asymmetry discovered in 2004 [6]. Those previous studies showed that the brain state can be detected and analyzed using EEG measurement and the machine learning algorithm; however, it is still hard to satisfy both the performance and explanation of cognition evaluation.

The study of cognitive load has been covered in previous studies [7,8]. The brain state is regarded as the comparison within individuals which is often related to feelings. Nonetheless, comparing cognitive abilities among individuals is difficult to analyze due to the diversity of brains in different subjects. In 2015, a study identified individuals using patterns of brain connectivity [7]. The result indicated that one’s brain connectivity can be treated as the structure of the cerebrum, which was unique to distinguishing subjects. One’s cognitive behavior can also be predicted by considering connectivity one’s brain network. This discovery proved the assumption concerning human brain diversity and the possibility to predict human behaviors. Another study in 2022 focused on the correlation between brain connectivity and EEG signals [8]. It was found that the base signal in EEG is shared across tasks and time, which can serve as a personal identifier. The base signals in EEG were coined “mind prints”, which originated from the uniqueness of individual brain networks. Furthermore, Abbasi et al. [9] showed through experimentation and clinical research that the severity of neuronal damage is also correlated with changes in EEG after hypoxic-ischemia. EEG signals are influenced by the age and severity of neuronal loss. Combining the above studies, it is possible to predict one’s working memory ability using EEG analysis.

Working memory processing is an important part of cognition which involves short-term memory and thinking. The evaluation of working memory ability has seldom been addressed in previous research. A study in 2020 predicted working memory ability based on EEG by functional data analysis [10]. Least absolute shrinkage and selection operator (LASSO) was utilized to select features and a linear regression model was constructed to predict working memory performance. The article claimed that there were no existing methods for predicting working memory ability. Another study in 2022 predicted a working memory performance based on the spatio-temporal features [11]. The result indicated that the spatial and temporal characteristics emerged during the working memory process, and that the working memory performance can be predicted using regularized linear discriminant analysis (RLDA). There was a correlation between the real performance and model prediction. These previous studies, however, shared the same defects. First, the error between prediction and real performance was not discussed. It is possible to predict working memory ability with more exact outputs. Second, insufficient regression analysis was conducted in both studies due to the lack of train–test validation. This was not compared with other regression models. These two main defects are the barriers to evaluating working memory ability.

In this research, we will focus on the entire evaluation of one’s working memory ability based on EEG analysis. The mechanism of working memory processing was discovered in terms of neurological aspects, looking further into the characteristics of EEG signals in different brain regions and states. On the other hand, the fuzzy support vector regression (FSVR) models are designed using the discovery from the mechanism of working memory processing and compared to other machine learning algorithms. The discussion of working memory processing from the EEG aspect will be covered in the whole article. The main contributions are as follows:Two propositions for the working memory model are described in terms of EEG analysis.The FSVR approach is proposed to evaluate working memory ability based on EEG signals.

## 2. Materials

The preparations of materials are demonstrated in this section. Dataset 1 was acquired with the 10-10 system Neuroscan device by the National Taiwan University Robotics Lab. We used this dataset as the main material to propose the working memory model and the algorithms. Dataset 2 is an open source dataset available since 2022 created by the University of Salford (UK), Aston University (UK) and University College Dublin (Ireland) [12]. This dataset was used to confirm that another algorithm with fuzzy concepts also performs better than existing approaches in other types of working memory ability. Although the two datasets involve the working memory performance, dataset 1 focuses on the capacity of working memory, and dataset 2 refers to the recalling speed. The profiles of these two datasets are shown in Table 1.

In dataset 1, an experiment comprised 20 subjects, 17 males and 3 females, between the ages of 22 and 24 with Bachelor’s degrees as their minimum level of education. To ensure that their mental states were relaxed and without fatigue, the subjects were instructed to sleep sufficiently the previous night and they rested for about 10 min before the experiment. The electrodes were selected according to their corresponding brain areas, which are located in the prefrontal lobe (Fpz), frontal lobe (Fz, F7, F8), temporal lobe (T7, T8), parietal lobe (Cz, Pz), and the occipital lobe (Poz, Oz). These 10 electrodes fully cover the entire cerebrum, and the analysis of EEG features is separated among these regions.

### 2.1. Measuring Procedure

The main objective of the experiment was to measure the EEG signals of the subjects and the time spent in each task. The whole procedure was divided into three sessions. In session 1, the subjects were asked to stare at the whiteboard for 1 min and think of nothing. The data recorded in this session constituted the brain waves in resting state. In session 2, the subjects conducted the memory task with 8 numbers. In session 3, the subjects conducted the memory task with 12 tokens. The subjects could interact with a gamer interface and each session was conducted 3 times. In the memory task, the subjects were asked to find the pair of cards with the same numbers or tokens. If the cards with the same items were chosen, these two cards disappeared. If cards with different items were chosen, these remained on the screen. The more accurate the subjects remembered, the faster the task was completed. Eventually, the time consumption was defined as the cognitive performance. A picture of the task is shown in Figure 1.

After the experiment, the cognitive performance and the EEG signals were recorded for working memory prediction. The cognitive performance measured using time was used to make a *Z*-score, which is the standard representation of cognitive ability and is the same as the Wechsler Adult Intelligence Scale. Since one’s working memory process is regarded as a part of human intelligence, one’s working memory performance was denoted by a *Z*-score as shown in Equation (Equation 1), which represents their ranking compared to the group. *x* is the inverse of the time that a subject takes in the measurement, μ is the average and σ is the standard deviation.
(1)$$z=\frac{x-\mu }{\sigma }$$

### 2.2. Working Memory and Dementia

Nowadays, subjects’ basic cognition is assessed by scales. The Wechsler Intelligence Scale (WMS) is utilized to measure different types of memory [13]. The Cognitive Abilities Screening Instrument (CASI) is good for diagnosing Alzheimer’s disease (AD) and mild cognitive impairment, but it is not suitable for highly educated subjects [14]. The Montreal Cognitive Assessment (MoCA) is a fast tool for detecting AD in patients composed of 30 questions measuring executive functions and multiple cognitive domains, as shown in the Appendix A, which represents the general cognitive ability of subjects [15]. The correlation between dementia and cognitive abilities has been confirmed. From the same scales mentioned above, general cognitive ability can be divided into several common parts, as shown in Table 2. In this study, we focus on working memory ability through a card pairing task, which mainly depends on anterograde space memory ability, anterograde word/image memory ability and the ability to focus. These three parts of brain function are the parts of basic cognition. To assess the working memory ability, we referred to the design of WMS, wherein the performance is denoted as a *Z*-score for the comparison among subjects. In our measurement, working memory ability is represented by a *Z*-score. People living with AD constitute 1.34% of the total population. If older subjects have working memory ability scores lower than −2, then there is a high risk that the subject has a mild cognitive impairment. Working memory performance can be treated as an index to rapidly predict dementia.

## 3. Methods

This section demonstrates the algorithms used in data processing and analysis. Using feature extraction, the EEG signals are integrated into the vector of features which includes the characteristics of working memory processing. The knowledge-based fuzzy SVR (FSVR) will be introduced, and the multi-linear SVR with fuzzy C-mean (FCM) clustering is also utilized to construct prediction models. The flowchart representing the FSVR design and process is shown in Figure 2.

### 3.1. Feature Extraction

The EEG signals are preprocessed for cleaner data. EEG signals are easily affected by muscle movement and eye blinks [4], and the loss of signals occurs due to accidents in the measuring procedure. The purpose of artifact removal is to eliminate abnormal signals and bad channels. After that, automatic preprocessing is designed for noise reduction. The low-frequency peak noise is removed by setting the threshold. The signal is re-referenced to the average, and the mean filter is applied to smooth the high-frequency noise. The whole procedure of signal preprocessing is shown in Figure 3.

After preprocessing, the fast Fourier transform is taken to calculate the band power. The features extracted from signals under specific spectra are called the band power ratio (BPratio), which is defined in Equations (Equation 2)–(Equation 4), as derived from our previous research [16]. The meaning of the band power ratio indicates the intensity of certain waves. In this research, we focus on alpha, beta and gamma waves since these three waves are often related to the conscious brain state. The alpha pattern represents the relaxed state, the beta pattern represents the thinking state, and the gamma pattern represents the focused state. The characteristics of EEG signals of working memory processing will be denoted by these three patterns. With 3 band power ratios and 10 channels, a feature vector consists of 30 components.
(2)$${BPratio}_{\alpha }=\frac{{BP}_{8–13\mathrm{Hz}}}{{BP}_{1–44\mathrm{Hz}}}$$
(3)$${BPratio}_{\beta }=\frac{{BP}_{13–30\mathrm{Hz}}}{{BP}_{1–44\mathrm{Hz}}}$$
(4)$${BPratio}_{\gamma }=\frac{{BP}_{30–44\mathrm{Hz}}}{{BP}_{1–44\mathrm{Hz}}}$$

### 3.2. Knowledge-Based Fuzzy SVR

Support vector regression (SVR) has been applied to many regressive problems to predict a continuous output [17]. Solved using the Lagrange multiplier, n support vectors $\alpha_{i}$ and $\alpha_{i}^{*}\;$ were found during the training process, which expanded the function *f*(*x*) with bias b, as shown in Equation (Equation 5). The kernel $k\left(x_{i},x\right)$ is often chosen as a Gaussian function for normal distribution data. In this research, SVR is treated as predictor of the feature vector.
(5)$$f\left(x\right)=\sum_{k=1}^{n}\left(\alpha_{i}-\alpha_{i}^{*}\right)k\left(x_{i},x\right)+b$$

Knowledge-based fuzzy SVR (FSVR) is a combination of the SVR regression part and the fuzzy logic part. It is designed based on the working memory model proposed in this research, which will be discussed in the next section. The procedure of designing the fuzzy logic part is the same as that of the traditional fuzzy logic controller. The BPratios of the alpha, beta and gamma waves from the same channel are transformed by the *Z*-score and serve as the input. As shown in Figure 4, the fuzzy membership function is designed as triangular function, and the relation between the input and the output can be visualized as the control surface, as shown in Figure 5. “Alpha” is the input of the *Z*-score ${BPratio}_{\alpha }$, “Beta” is the input of the *Z*-score ${BPratio}_{\beta }$, and “Output” denotes the outputs of control surface.

The fuzzy logic part of knowledge-based FSVR serves as a feature transformer which changes the original feature vector with a length of 30 into the new feature vector with a length of 10. These 10 new features stand for the degree satisfying the rule of a working memory model in each channel. The proposed model balances the requirements of explanation and performance, which modifies the features by fuzzy rules with scientific knowledge and achieves a high prediction performance.

### 3.3. Multi-Linear SVR with Fuzzy C-Mean Clustering

Multi-linear SVR with FCM clustering is a modified fuzzy SVR regressing data by multiple SVR models proposed in [18]. Given fixed amounts of clusters, the centroid of each cluster is defined in Equation (Equation 6), where w(x) is the fuzzy membership degree of the *x* corresponding to the centroid $c_{k}$. Parameter m represents the fuzzified degree of the clusters. To determine the proper position of centroids, the FCM aims to minimize the objective function, as shown in Equation (Equation 7).
(6)$$c_{k}=\frac{\Sigma_{x}w_{\left(x\right)}^{m}x}{\Sigma_{x}w_{\left(x\right)}^{m}}$$
(7)$$J_{\left(w,C\right)}=\sum_{i=1}^{n}\sum_{j=1}^{C}{w_{ij}^{m}∥x_{i}-C_{j}∥}^{2}$$

Using FCM clustering, each of the clusters has one regression model. The input feature vector is projected onto the fuzzy membership degree space of multi-clusters, and the output is the fusion of multi-linear SVR model. In this research, multi-linear SVR with fuzzy C-mean clustering is implemented to predict the working memory performance. The data are first regressed by SVR with a Gaussian kernel to set up the baseline. If the predicted error of the single data is less than the threshold (<0.5), this point will be included to the clustering set which decides the final model. The model is trained under certain amounts of clusters, which comprise hyperparameters needing to be tuned (integer 2 to 10). Input data are transformed into the fuzzy membership degree and finally construct the whole model. The results compared to other algorithms will be discussed in the next section.

## 4. Results

To fully investigate the working memory process, statistical analysis was applied to dataset 1, as required in our experiments and measurements. EEG signals demonstrate several characteristics of the working memory process based on which the working memory ability can be more accurately predicted with an explainable model.

### 4.1. Characteristics of EEG Signals

The positions of EEG channels are related to the functional division of the cerebral cortex. The superior frontal gyrus was located in the prefrontal lobe (Fpz, Fz), corresponding to the high cognitive function including decision making and planning. cuneus is located in the occipital lobe, corresponding to the processing and integration of visual information. These two brain regions are related to the working memory task in our experiments. The following analysis will be focused on the channels in the superior frontal gyrus, cuneus and full brain region.

Figure 6 shows an example of the EEG signals of subjects performing the working memory processing task (right) compared to subjects in the resting state (left). The energy of the alpha wave decreases, which is referred to as the relaxed state of the brain. The beta and gamma waves increase, which are referred to as the thinking and focusing states, respectively. Based on the tendency of changes in energy, we proposed Proposition 1 for the working memory task.

**Proposition** **1.** 
*The alpha waveform decreases, while the beta and gamma waveforms increase during the working memory process.*


This requires the functions of thinking and focusing, rather than relaxing. As shown in Table 3, the probability of satisfying Proposition 1 is higher than 80% for the alpha waveforms in the brain region in all subjects, but higher than 70% for the beta and gamma waveforms in only the cuneus and full region. The lack of probability of Proposition 1 being satisfied in the superior frontal gyrus is probably because the working memory process is a fundamental cognitive activity, and the superior frontal gyrus performs higher cognitive functions. With the explainable exception, the characteristics of EEG signals in different working memory task and resting state have been discovered.

To discover the relation between working memory ability and waveform energy, a correlation analysis was conducted. The working memory performance was defined as the inverse ratio of time consumption. Since Proposition 1 was not satisfied in the superior frontal gyrus, the following analysis will be focused on the cuneus and full brain region. The relation between performance and wave form energy shows the tendency in the full region. There is a negative relation in Figure 7a, and a positive relation in Figure 7b,c. The relation between performance and waveform energy shows the tendency in the cuneus. There is a negative relation in Figure 8a, and a positive relation in Figure 8b,c. Based on the tendency, we propose Proposition 2.

**Proposition** **2.** 
*The energy of the alpha waveform shows negative effects on working memory performance while beta and gamma waveforms have positive effects on the working memory performance.*


As shown in Table 4, the result in the picture task is significant, which supports Proposition 2. The lack of satisfaction for the word task is probably due to the defect of the task in experiments. There are 8 pairs in word memory task, which is less than the 12 pairs in the picture memory task, and not challenging enough for subjects with significant differences in their abilities. The performance in the word task was strongly affected by the random initials of the card positions. Based on the evidence, Proposition 2 is confirmed with the explainable exception.

Summarizing the two propositions and the discovered EEG signal characteristics, we propose a working memory model for waveform aspects with two equations. Equation (Equation 8) shows the energy change in the working memory process from resting state. ${BPratio}_{\alpha }$ decreases with ${BPratio}_{\beta }$ and ${BPratio}_{\gamma }$ increases. Equation (Equation 9) describes the effects of the waveform energy on the working memory performance. The lower the energy of the alpha waveform, the higher the energies of the beta and gamma waveforms and the better the subject performs in the working memory task. Compared to the entire group, the performance of a task can determine the working memory ability of the subject.
(8)$$\Delta {BPratio}_{\alpha }<0\;,\Delta {BPratio}_{\beta }>0\;,\Delta {BPratio}_{\gamma }>0$$
(9)$$Performance\propto \frac{{BPratio}_{\alpha }}{{BPratio}_{\beta }\;{BPratio}_{\gamma }}$$

### 4.2. Zen Meditation and Working Memory Process

Zen meditation is a traditional Buddhist art which is often related to the effects of releasing mental stress and relaxing. A previous study [19] divided Zen meditation into five scenarios. In scenario D, the meditator experiences a bright (surrounded by sacred light) and fully relaxed meditation in the test. The alpha activities also emerge at this moment. The results of the alpha waveform during Zen meditation show a relaxed brain state, which is preferred during the working memory task. In another previous study in 2006 [20], the author hypothesized that Zen meditation results in decreased cognitive involvement and decreased mental activity, which is the opposite to the brain state during the working memory task. Combining our findings with those of previous articles, the change in alpha energy from the resting state reflects different brain activities in Zen meditation and the working memory task. As shown in Table 5, the change in alpha energy reflects the degree of cognition deployed. The effects of Zen meditation on alpha waveform energy are the opposite to those of the working memory task.

### 4.3. Prediction of Working Memory Ability

The working memory abilities of subjects in dataset 1 are predicted using the method proposed in this research. The models built by different approaches are shown in Table 6. LASSO was utilized in previous research [10] with different feature extraction methods. Knowledge-based FSVR is designed using the working memory model of the waveform aspect, which is proposed in Section 4.1, and fuzzy logic regression is the fuzzy logic part of knowledge-based FSVR. In dataset 1, the results show that knowledge-based FSVR and SVR with FCM clustering perform better than other approaches, especially knowledge-based FSVR, which achieves the least mean square error. To verify the performance on different types of memory task, dataset 2 is utilized to confirm the advantages of SVR with FCM clustering. It achieves the least mean square error compared to other approaches. Knowledge-based FSVR and fuzzy logic regression are not available on dataset 2 as their fuzzy inferences are designed based on the discovery in dataset 1. The definition of performance in dataset 2 is the recalling speed, and it is not equivalent to our defined performance in dataset 1. This makes knowledge-based FSVR incompatible with dataset 2. Through two different datasets, the performances of the two approaches proposed in this research are demonstrated.

## 5. Conclusions

In this research, the main objective was to discover the working memory ability in terms of the EEG aspect, and we designed the experiment for the measurement. The working memory model of the EEG aspect was proposed with a correlation analysis, which discovered that the usage of the working memory process involves the energy change in waveform energy, and the working memory performance was related to the change in waveform energy. The model was supported by the significant *p*-value from statistics correlation analysis. Based on this model, the working memory ability was predicted and the brain state during the working memory process was discovered. It is proven that one’s working memory ability can be evaluated from EEG signals using knowledge-based FSVR and multi-linear SVR with FCM clustering. The proposed method not only balances the explanation and performance of model, but also goes through train–test validation on the working memory EEG dataset compared to other regression models. The results demonstrate that knowledge-based FSVR achieves the best performance in the regression problem of dataset 1, and multi-linear SVR with FCM clustering achieves the best performance in the regression problem in dataset 2. Our study demonstrates an example of cognitive function evaluation by EEG measurement. In the future, other higher cognitive abilities can be evaluated based on the proposed methods.

## Figures and Tables

**Figure 1 sensors-23-08246-f001:**
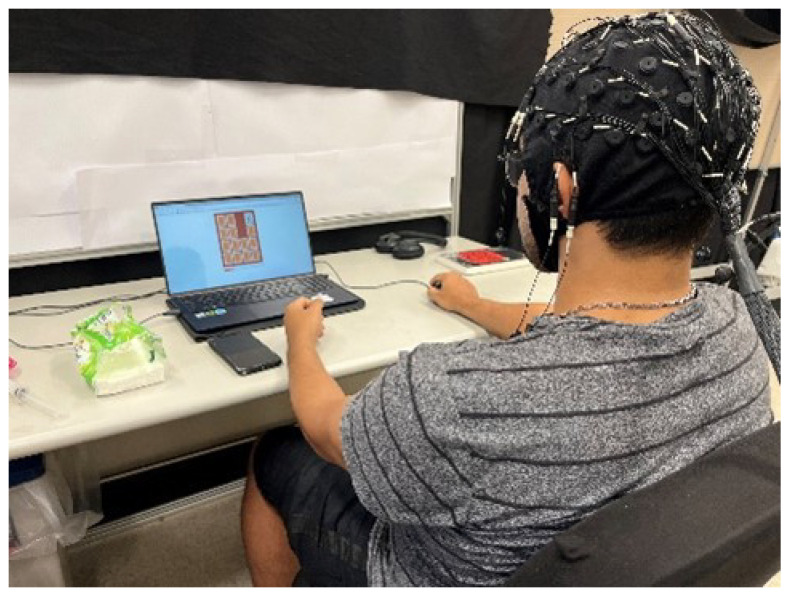
Measurement during working memory task.

**Figure 2 sensors-23-08246-f002:**
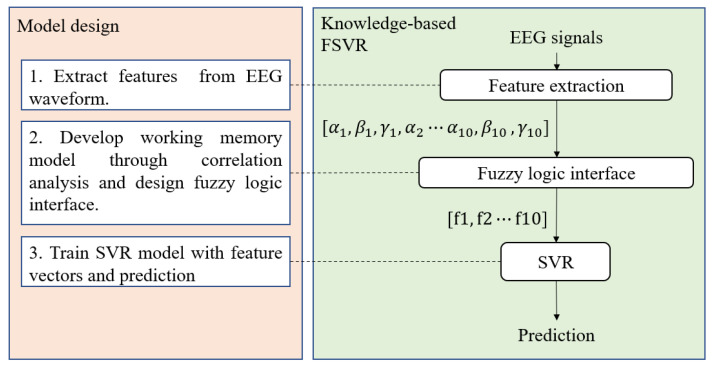
Knowledge-based FSVR.

**Figure 3 sensors-23-08246-f003:**
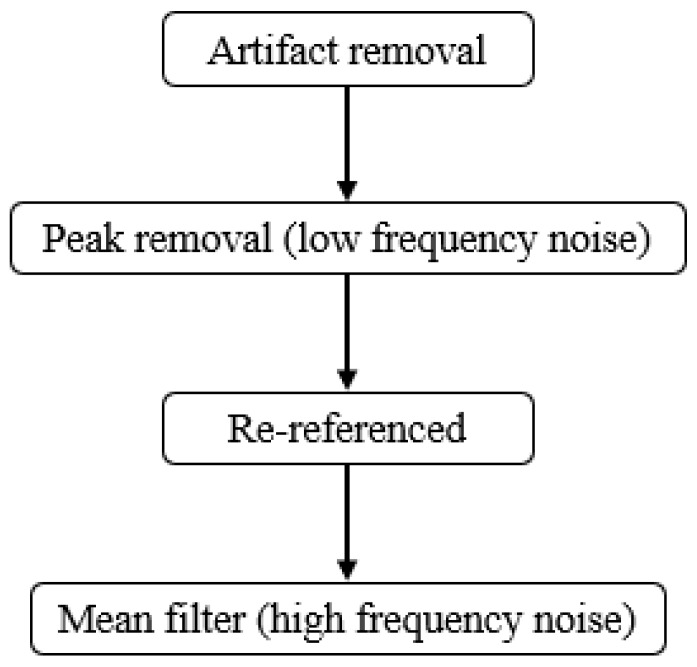
Preprocessing of EEG signals.

**Figure 4 sensors-23-08246-f004:**
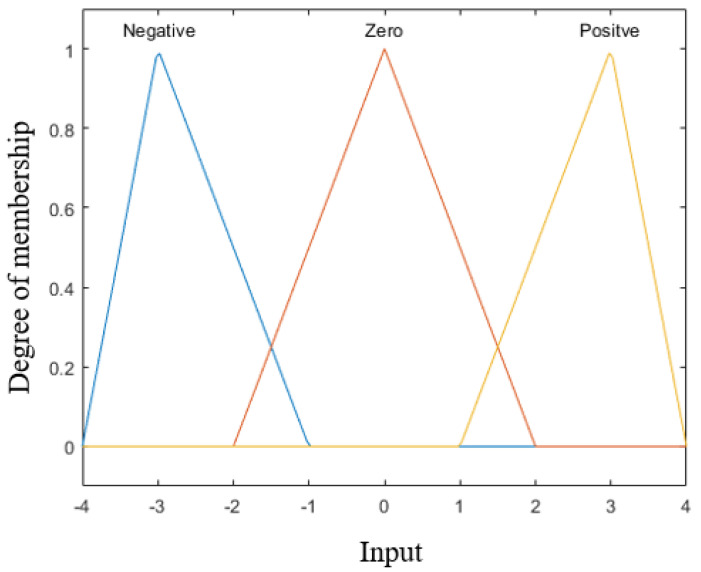
Membership function.

**Figure 5 sensors-23-08246-f005:**
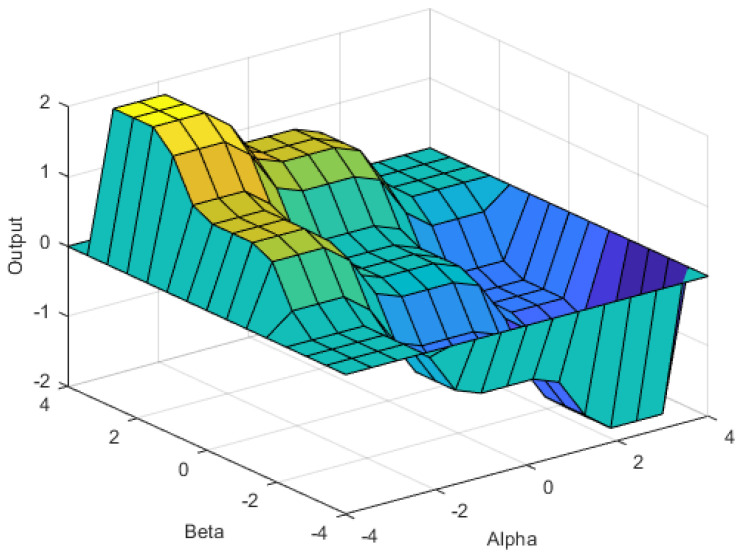
Fuzzy control surface of alpha and beta.

**Figure 6 sensors-23-08246-f006:**
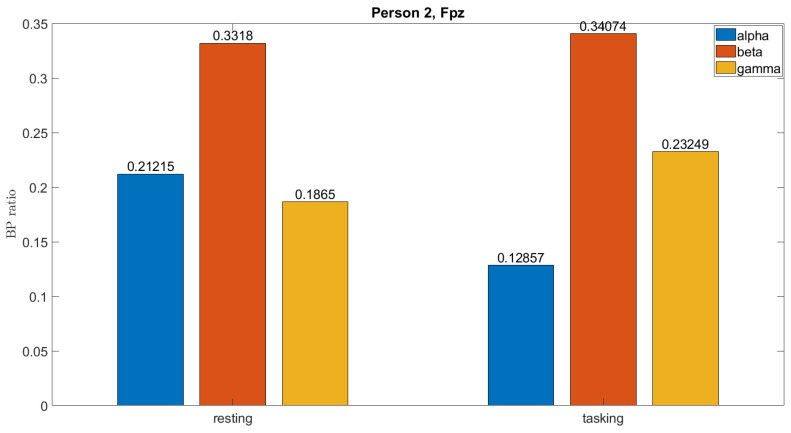
Waveform changes during the working memory process.

**Figure 7 sensors-23-08246-f007:**
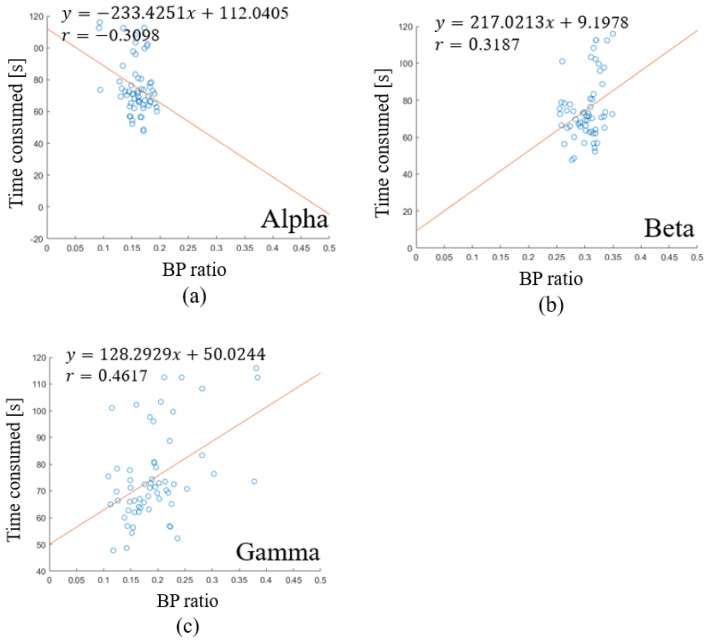
Linear regression of waveform energy and time consumed in full region. (**a**) the relation between alpha band power ratio and time consumed. (**b**) the relation between beta band power ratio and time consumed. (**c**) the relation between gamma band power ratio and time consumed.

**Figure 8 sensors-23-08246-f008:**
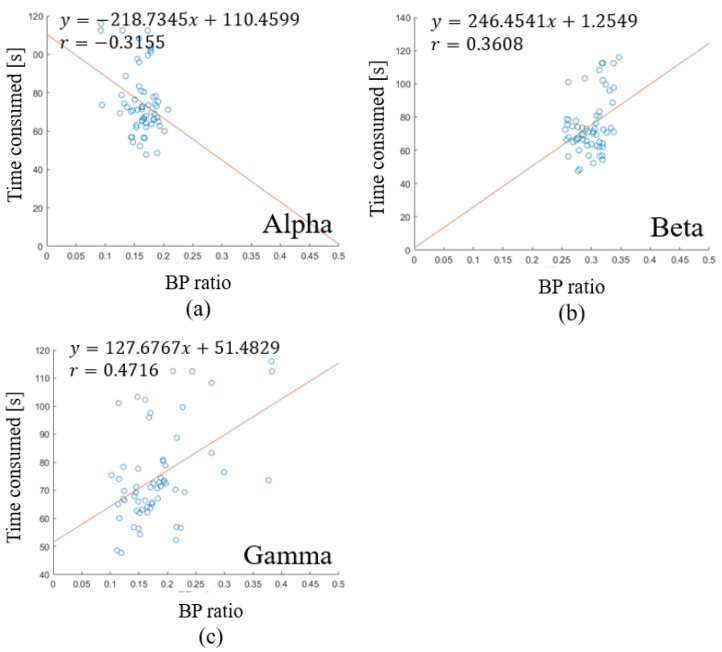
Linear regression of waveform energy and time consumed in cuneus: (**a**) the relation between alpha band power ratio and time consumed; (**b**) the relation between beta band power ratio and time consumed; (**c**) the relation between gamma band power ratio and time consumed.

**Table 1 sensors-23-08246-t001:** Profile of two datasets.

	Dataset 1	Dataset 2
Date	August 2022	January 2022
Subjects	20	47
Sessions	2	2
Trials	3 + 1 resting state	112
Channels	10	61
Sampling rate	1000/s	500/s
Data size	120	10,528
Label	Time consumption of working memory task	Time consumption of memory recalling

**Table 2 sensors-23-08246-t002:** Basic cognition.

Categories	Components
Anterograde memory	Linguistic logic
	Space *
	Facial recognition
	Word and image *
Sense	Time
	Position
Focusing *	
Oral fluency	
Mind operation	Word reconstruction
	Drawing mimic

* Types of cognition related to working memory ability.

**Table 3 sensors-23-08246-t003:** Percentages of satisfying Proposition 1.

Brain Region	Alpha	Beta	Gamma	Task
Superior frontal gyrus	80 *	55	65	Word
	85 *	50	55	Picture
Cuneus	95 *	70 *	90 *	Word
	90 *	75 *	85 *	Picture
Full region	85 *	75 *	85 *	Word
	85 *	85 *	80 *	Picture

* High tendency to satisfy Proposition 1.

**Table 4 sensors-23-08246-t004:** *p*-value correlation analysis.

Brain Region	Alpha	Beta	Gamma	Task
Superior frontal gyrus	0.317	0.983	0.686	Word
	0.0141 *	0.00463 *	0.000143 *	Picture
Cuneus	0.231	0.619	0.718	Word
	0.0160 *	0.0131 *	0.0002 *	Picture

* is the *p*-value < 0.05 for Proposition 2.

**Table 5 sensors-23-08246-t005:** Comparison between working memory task and Zen meditation.

	Working Memory Task	Zen Meditation
Alpha energy	Decrease	Increase
Cognition deployed	Increase	Decrease

**Table 6 sensors-23-08246-t006:** Mean square errors of models in terms of *Z*-score.

Approaches	Dataset 1	Dataset 2
LASSO	2.85	1.9
SVR	0.99	1.17
Fuzzy logic regression	1.03	1.36
SVR with FCM clustering	0.76	1.111
Knowledge-based FSVR *	0.60	Not available

* is the proposed model.

## Data Availability

Not applicable.

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
