# Peer review of "Working Memory Ability Evaluation Based on Fuzzy Support Vector Regression"

_sensors, 2023, doi:10.3390/s23198246_

Round 1
Reviewer 1 Report (Previous Reviewer 2)
The authors have addressed my comments and provided clarifications where needed. Their responses and the subsequent revisions to the manuscript have enhanced its quality and resolved the concerns I initially raised.
Author Response
Thank you for the comments.
Reviewer 2 Report (New Reviewer)
Reviewer’s Report on the manuscript entitled:
Working Memory Ability Evaluation based on Fuzzy Support Vector Regression
The authors proposed an approach to evaluate working memory ability with Electroencephalography (EEG) analysis. They showed that working memory ability can be evaluated from EEG signals by using Knowledge-based FSVR and muti-linear SVR with FCM clustering. The topic and results are generally interesting, but the presentation and literature review should be improved. Below please see my comments.
The literature review is poor and should be improved. At least 10 recent articles about EEG, artifacts in EEG, applications in working memory, and machine learning models for processing EEG signals should be included.
Lines 24 and 32. Please include the following article for LSTM and KNN:
https://doi.org/10.1109/JSEN.2023.3237383
Line 144. The article above can also be included here as it presents artifact detection and removal techniques from EEG signals. The sources of artifacts in EEG signals include eyeblink, muscle movement, etc. Please describe these briefly.
Lines 51-66. Abbasi et al. [https://doi.org/10.3390/signals4030034] showed though experimentation and clinical research that severity of neuronal damage is also correlated with changes in EEG after hypoxic-ischemia. Please also elaborate briefly on in Introduction.
Lines 68-74. It is not clear what the main contributions are. Please highlight them preferably using bullet points.
Line 83. Grammar issue.
Figure 2. Pre-processing including artifact detection and removal is also important to be included in the flowchart.
Equations (2)-(4). I suggest making the symbology better. The subscripts can be shortened with appropriate format/style.
What is “b” in Equation (5)?
Figure 4. What are “A”, “B”, and “U” in the labels? Di you mean Alpha, Beta, etc.? Please write the full names of axis labels.
The legend of Figure 5 should be inside not overlapping. In addition, this figure has a very large font size.
Please add the p-values of the estimated slopes in Figure 7.
Table 3. The * values do not seem to follow Proposition I. For example, see Cuneus 90 for words and 85 for pictures. Please check and make necessary changes.
The font size of axis labels in Figures 6 and 7 are too small and should be enlarged. Generally, the font size of text and numbers in all the figures should be the same as the font size of the figure caption, so it is consistent, nice, professional, and readable.
Line 245. “Equation (9)” not “The equation (9)”.
Please highlight the limitation of this study better at the end of the manuscript.
Thank you!
Regards,
There are many grammar/style issues that should be checked and corrected.
Author Response
The reply is attached below.

Round 2
Reviewer 2 Report (New Reviewer)
Dear authors,
Thank you for addressing my comments and improving your manuscript. Please make sure the font size of tests and numbers in your figures are consistent and they have a resolution of at least 300 dpi to make your work look more professional.
Please insert the page numbers for some of the articles in the reference list.
Best regards,
Some minor editorial typos/punctuation corrections. Please carefully proofread the manuscript.
This manuscript is a resubmission of an earlier submission. The following is a list of the peer review reports and author responses from that submission.
Round 1
Reviewer 1 Report
Overall content of the article just fair.
Abstract:
It is suggested to author to provide the accuracy results of the proposed method (SVR, FCM and FSVR).
1. Introduction:
There are lack of citations of the references that are related to the research. There are only 7 articles are cited in this section. It is suggested to author to provide at least 15 references in this section. However, the citation are in chronological order. Some references are recent references (published in year 2022).
2. Materials
i) The measuring procedure is not comprehensive and clear. It is suggested to describe the EEG instruments that are employed in the study in term of sampling frequency and the software that associated with the EEG instruments. Author needs to explain the duration of the measurement for session 2 and 3. Besides, author need to explain rationale of using 2 different datasets with different number of subjects and EEG's channels are employed in the study. Author also need to explain about the meaning of token.
ii) Since the experimental work involved humans as experimental subject, the work ethical approval should be provided. There is no work ethical approval found in the article.
iii) Why Z-score technique is used to analyze the cognitive performance?
iv) It is suggested to author to share the example of the 30 questions of the MoCA in the appendix section of the article.
3. Methods
It is suggested to author to start this section with the overall process flowchart or block diagram of the study. Arrange the presentation of this section according to process flowchart. The rest of the content of this section is good.
4. Results
i) The graph of EEG signals during tasking of working memory and resting-state for subject 2 as shown in Figure 5 is good. However, it is suggested to author to share graph of other subject as for comparison. Why only BP ratio at FPz are shared and discussed in the article. How about analysis of the EEG at another channel or location? It is better to share results from all channels and subjects.
ii) The results that are displayed by Figure 6 and 7 are good.
iii) The results shown in Table 4, 5 and 6 are also good. However, it is suggested to author to represent results in Table 6 in term of accuracy as well.
iv) Results of the Performance task, equation 8 is not presented.
5. Conclusion
Overall content of the section just fair and quite short. It is suggested to author to represent this section in term of accuracy and correlation results of the proposed method.
References
i) The number of references which are only 16 are too short for article or journal. It is suggested to author to cite from more than 30 references.
ii) Some references are good and very recent (published in year 2022).
Author Response
Overall content of the article just fair.
Abstract:
It is suggested to author to provide the accuracy results of the proposed method (SVR, FCM and FSVR).
The contents will be modified in the latest version.
- Introduction:
There are lack of citations of the references that are related to the research. There are only 7 articles are cited in this section. It is suggested to author to provide at least 15 references in this section. However, the citation are in chronological order. Some references are recent references (published in year 2022).
The issue of predicting working memory ability were seldom discussed in previous research. The references include the related topic, such as cognitive load, and other articles that support our discovery.
- Materials
- i) The measuring procedure is not comprehensive and clear. It is suggested to describe the EEG instruments that are employed in the study in term of sampling frequency and the software that associated with the EEG instruments. Author needs to explain the duration of the measurement for session 2 and 3. Besides, author need to explain rationale of using 2 different datasets with different number of subjects and EEG's channels are employed in the study. Author also need to explain about the meaning of token.
The contents will be modified in the latest version. The measuring specification were already noted in Table 1.
- ii) Since the experimental work involved humans as experimental subject, the work ethical approval should be provided. There is no work ethical approval found in the article.
We are still working on it.
iii) Why Z-score technique is used to analyze the cognitive performance?
The contents will be modified in the latest version.
- iv) It is suggested to author to share the example of the 30 questions of the MoCA in the appendix section of the article.
The contents will be modified in the latest version.
- Methods
It is suggested to author to start this section with the overall process flowchart or block diagram of the study. Arrange the presentation of this section according to process flowchart. The rest of the content of this section is good.
The flowchart with design and signal process was shown in Figure 4, covering from section 3A and 3B. section 3C is another type of algorithm discussed independently.
- Results
- i) The graph of EEG signals during tasking of working memory and resting-state for subject 2 as shown in Figure 5 is good. However, it is suggested to author to share graph of other subject as for comparison. Why only BP ratio at FPz are shared and discussed in the article. How about analysis of the EEG at another channel or location? It is better to share results from all channels and subjects.
We analyze the BP ratio of 10 channels within 20 subjects, and the result were shown in Table 3. Among totally 200 analyses, it is suitable to shows the example of one of the analysis rather than all of them.
- ii) The results that are displayed by Figure 6 and 7 are good.
Thanks for the compliment.
iii) The results shown in Table 4, 5 and 6 are also good. However, it is suggested to author to represent results in Table 6 in term of accuracy as well.
In regression problem, mean square error is served as the performance, which is equivalent to the accuracy in classification problem. We think it is proper to use mean square error in Table 6.
- iv) Results of the Performance task, equation 8 is not presented.
The equation normally display in our PDF and LaTex file. We are not sure what is the problem.
- Conclusion
Overall content of the section just fair and quite short. It is suggested to author to represent this section in term of accuracy and correlation results of the proposed method.
The contents will be modified in the latest version.
References
- The number of references which are only 16 are too short for article or journal. It is suggested to author to cite from more than 30 references.
The issue of predicting working memory ability were seldom discussed in previous research. The references include the related topic, such as cognitive load, and other articles that support our discovery.
- Some references are good and very recent (published in year 2022).
Thanks for the compliment.
Reviewer 2 Report
This study investigates the prediction of working memory abilities using EEG signals. The researchers designed experiments, proposed a working memory model based on correlation analysis, and demonstrated that EEG signals can effectively be used to evaluate working memory abilities using Knowledge-based FSVR and multi-linear SVR with FCM clustering. The proposed method, validated using a working memory EEG dataset, offers an effective balance between explanation and performance of the model. This study indicates potential for further research in cognitive neuroscience, suggesting that similar methodologies can be applied to evaluate other higher cognitive abilities.
Major comments:
1. The study primarily focuses on the alpha, beta, and gamma waveforms, but it would be interesting to see if other frequency bands, like theta or delta, also have some correlation with working memory.
2. Although the correlation between EEG waveforms and working memory is well explored, the causation is not clearly demonstrated. More sophisticated statistical analyses or experimental designs could help address this.
3. The paper discusses the results of Zen meditation and its effects on alpha waveform energy. However, it would be beneficial to clarify the connection between this and working memory more clearly.
4. More diverse datasets, especially ones with varying working memory tasks, could be employed to further validate the robustness of the proposed models.
5. The authors may consider including a discussion section where they interpret their findings in light of existing literature. This would help the reader understand how their study fits into the broader context of cognitive neuroscience.
6. The paper might benefit from a more detailed conclusion, elaborating on the implications of the study and outlining potential avenues for future research.
7. While the study presents some intriguing findings, there is a lack of detail on the actual methodology employed. Future iterations of the study could benefit from a more comprehensive explanation of the experimental setup and data analysis procedures.
The quality of the English language in the provided text is generally good, with mostly clear and comprehensible sentence structures. However, there are some minor grammatical errors and awkward phrasings.
Author Response
This study investigates the prediction of working memory abilities using EEG signals. The researchers designed experiments, proposed a working memory model based on correlation analysis, and demonstrated that EEG signals can effectively be used to evaluate working memory abilities using Knowledge-based FSVR and multi-linear SVR with FCM clustering. The proposed method, validated using a working memory EEG dataset, offers an effective balance between explanation and performance of the model. This study indicates potential for further research in cognitive neuroscience, suggesting that similar methodologies can be applied to evaluate other higher cognitive abilities.
Major comments:
- The study primarily focuses on the alpha, beta, and gamma waveforms, but it would be interesting to see if other frequency bands, like theta or delta, also have some correlation with working memory.
According to the neuroscience aspect, delta and theta waveforms involve the sleeping or drowsy brain states, which are not satisfy the condition of our experiment, and we did not discover the significant correlation with working memory process.
- Although the correlation between EEG waveforms and working memory is well explored, the causation is not clearly demonstrated. More sophisticated statistical analyses or experimental designs could help address this.
Thanks for the advice. We are still working on the research of cognition and EEG waveforms. A more complete describe will be proposed in the future works.
- The paper discusses the results of Zen meditation and its effects on alpha waveform energy. However, it would be beneficial to clarify the connection between this and working memory more clearly.
The research of Zen meditation and its effects were found in the referenced article; however, after that research it was seldom discussed until nowadays. The knowledge limitation of Zen meditation makes it hard to be clarify with details. We will attempt to find out the inverse procedure of working memory process.
- More diverse datasets, especially ones with varying working memory tasks, could be employed to further validate the robustness of the proposed models.
The dataset 2 is used to validate the performance of the methods used in self-collected dataset 1.
- The authors may consider including a discussion section where they interpret their findings in light of existing literature. This would help the reader understand how their study fits into the broader context of cognitive neuroscience.
The previous study and the connection with context of cognitive neuroscience were describe in section 1. The relation between working memory and cognition were describe in section 2B.
- The paper might benefit from a more detailed conclusion, elaborating on the implications of the study and outlining potential avenues for future research.
The contents will be modified in the latest version.
- While the study presents some intriguing findings, there is a lack of detail on the actual methodology employed. Future iterations of the study could benefit from a more comprehensive explanation of the experimental setup and data analysis procedures.
The details of datasets and experimental settings were shown in section 2. The proposed methods were explained in section 3. The findings were discussed in section 4.
Comments on the Quality of English Language
The quality of the English language in the provided text is generally good, with mostly clear and comprehensible sentence structures. However, there are some minor grammatical errors and awkward phrasings.
The contents will be modified in the latest version.
Round 2
Reviewer 1 Report
Overall, the content of the paper much improve than before especially on Abstract section, Introduction sections & Materials section (section 2.2) where author had rephrased sentences or add new sentences. However, here are some comments that author might need consider;
Abstract section
i) It is suggested to clarify the word "the highest accuracy", maybe can mention it in value such as 90% accuracy.
Introduction section
i) Even though the content of this section look much better that before but not yet comprehensive. Thus, it is suggested to author to add more references on this section that related to the topic of the research. Currently, there are only 7 references are cited in that section.
Materials section
i) Any good justification to have 2 different sets of the Datasets? Are the datasets measured from the same EEG device? Besides, there are different sampling rate between the 2 datasets.
ii) Work ethical approval is NOT mentioned in this section. Since the experimental works employ human as an experimental subject, the work ethical approval must be obtained before conducting the experiment.
iii) Z-score is mentioned in section 2.2 and 2.3. However, the formula for Z-score is NOT mentioned in the article.
iv) In section 2.2 (line 115), author has mentioned about Appendix 1. However, Appendix 1 CANNOT be found in the article.
Methods section
i) This section supposed to start with process flowchart or block diagram of the study. However, there is NO flowchart or block diagram of the study in the article.
Result section
Overall, the result section of the article is good and well presented. However, the result of the Z-scores is NOT discussed in this section.
Conclusion section
The content of this section should tally with abstract and objective of the study. The summary results of the study should be mentioned in this section including BP ratios, mean square error, z-score, FCM, statistical parameter (P-value), linear regression of EEG bands energy.
Acknowledgments - Ok.
References - There are NOT ENOUGH references since the paper or article is going to be published in Journal.
Reviewer 2 Report
In general, few of my comments have been taken into account. There are issues that still should be addressed.
In the initial review, my recommendation was to 'Reconsider after major revision'. I had provided seven critical comments for the authors to address. Unfortunately, in their revised version, the authors only implemented minimal changes - they added or modified three brief sentences. From my perspective, this doesn't constitute a substantial response to the feedback provided.
This leads me to reconsider my previous decision and opt for a 'Reject' instead of 'Reconsider after major revision' for this second round. It seems clear to me that the authors didn't fully address the significant revisions suggested, which is vital for an article of this nature.
The quality of the English language in the provided text is generally good, with mostly clear and comprehensible sentence structures. However, there are some minor grammatical errors and awkward phrasings.